# *Brucella* spp. Contamination in Artisanal Unpasteurized Dairy Products: An Emerging Foodborne Threat in Tunisia

**DOI:** 10.3390/foods11152269

**Published:** 2022-07-29

**Authors:** Awatef Béjaoui, Ibtihel Ben Abdallah, Abderrazak Maaroufi

**Affiliations:** Group of Bacteriology and Biotechnology Development, Laboratory of Epidemiology and Veterinary Microbiology, Institut Pasteur de Tunis, University of Tunis El Manar (UTM), Tunis 1002, Tunisia; benabdallahibtihel25@gmail.com (I.B.A.); abderrazak.maaroufi@pasteur.tn (A.M.)

**Keywords:** *Brucella* spp., *B. abortus*, *B. melitensis*, raw milk, dairy products, foodborne pathogen, qPCR testing

## Abstract

Brucellosis is a worldwide zoonotic disease transmitted to humans, predominantly by the consumption of contaminated raw milk and dairy products. This study aimed to investigate the occurrence of *Brucella* spp. in 200 raw milk, ricotta, and artisan fresh cheese samples, collected from individual marketing points in four districts in Tunisia. Samples were analyzed for the presence of *Brucella* spp. by IS711-based real-time PCR assay. Positive samples were further analyzed by qPCR for *B. melitensis* and *B. abortus* species differentiation. The DNA of *Brucella* spp. was detected in 75% of the samples, *B. abortus* was detected in 31.3%, and *B. melitensis* was detected in 5.3% of positive samples. A percentage of 49.3% of samples co-harbored both species, while 14% of the *Brucella* spp. positive samples were not identified either as *B. abortus* or *B. melitensis*. High contamination rates were found in ricotta (86.2%), cheese (69.6%), and raw milk (72.5%) samples. The study is the first in Tunisia to assess the occurrence of *Brucella* spp. contamination in artisanal unpasteurized dairy products and showed high contamination rates. The detection of both *B. abortus* and *B. melitensis* highlights that zoonotic high-pathogen agent control remains a challenge for food safety and consumer health protection and could represent a serious emerging foodborne disease in Tunisia.

## 1. Introduction

Brucellosis is a major zoonosis caused by Gram-negative bacteria belonging to the genus *Brucella*. The World Health Organization (WHO) has classified the disease as one of the seven most “neglected zoonotic diseases” [1]. In the Middle East, Central Asia, north and east Africa, and Mediterranean countries, brucellosis is still endemic in humans and livestock. Political and socio-economic changes in this area resulting in impaired prophylactic surveillance systems and decreased border controls are believed to be the main reasons that make this region a hot spot for brucellosis [2]. At least 12 species of *Brucella* are currently known, where the main virulent species are *B. melitensis* followed by *B. abortus* and *B. suis*, which infect small ruminants, cattle, and swine, respectively [3]. These species are of particular importance in human health and livestock worldwide. Indeed, the disease causes clinical morbidity in humans and a considerable loss of productivity in animals. The disease in livestock mainly affects the reproductive tract and the udder, leading to severe losses, such as female abortion, infertility, and reduced milk production [4]. The excretion of the bacteria in milk is frequent and presents a serious risk for consumers of raw milk and its derivatives. In humans, the most common presentations of the disease in the acute phase are characterized by general malaise, including intermittent fever, headaches, myalgia, and arthralgia [5]. The infection can turn into a chronic disease and lead to severe affections with serious disabilities, such as hepatitis, endocarditis, and meningitis [6]. Endocarditis remains the principal cause of mortality if the disease is not adequately treated. Brucellosis is considered an occupational hazard for shepherds, slaughterhouse workers, veterinarians, dairy industry workers, and microbiology laboratory personnel [2]. The transmission of human brucellosis in rural populations occurs mainly through direct contact with infected animals, abortion products (placenta, fetuses, etc.), or from a contaminated environment, while the consumption of unpasteurized contaminated dairy products is the main way of contamination in the urban population [6,7]. The extension of animal industries and urbanization, poor hygienic measures in husbandry, and food handling practices in low-income countries are some of the reasons why brucellosis remains a public health concern [8]. Although there are no reliable data on the global burden of human brucellosis, an estimate of 5,000,000 cases per year is usually suggested, however, the real incidence is estimated to be 12,500,000, with about half of the cases being from foodborne origin [9].

The aim of this study, using molecular tools, was to estimate the occurrence of *Brucella* spp. contamination in raw dairy products, marketed at the retail level in the north of Tunisia, and to determine the distribution of *B. melitensis* and *B. abortus* to emphasize the hazard for consumers and to underline the risk of *Brucella* contamination as an emergent foodborne disease.

## 2. Materials and Methods

### 2.1. Samples Collection

The samples were collected from four districts in the north of Tunisia (Figure 1), which were located in the sub-humid (Bizerte and Beja) and semi-arid areas (Grand Tunis and Zaghouan). This area is characterized by a typical Mediterranean climate with hot summers of up to 22 °C, and precipitation occurring in the winter, ranging from 400 to 500 mm of rainfall per year [10]. Samples were purchased from 75 randomly selected unorganized retail marketing points for dairy products during the period from March to November 2019. A total of 200 samples of cow’s raw milk (*n* = 40), artisanal fresh cheese (*n* = 102), and ricotta (*n* = 58) were collected. The fresh cheese and ricotta samples were made from unpasteurized cow′s milk. All products were not packaged and had no indication that they had been inspected by any Tunisian organization involved in food safety. Each sample was bagged in a sterile bag and transported to the laboratory under refrigeration in a cool box.

### 2.2. DNA Extraction

From each cheese sample a total of about 10 g taken from different strata were manually and aseptically finely minced and homogenized, then an amount of about 30 mg was used for DNA extraction, while a volume of 100 µL was used for the milk samples. The DNA extraction was performed with the QIAamp DNA Mini Kit (QIAGEN, Hilden, Germany), according to the manufacturer’s instructions. As a negative control during the DNA extraction step, we used 100 µL of pathogen-free fetal bovine serum (FBS) (GibcoBRL, Paisley, UK).

The purified template DNAs were eluted with 100 µL of AE buffer (QIAGEN, Hilden, Germany) and conserved at −20 °C.

### 2.3. qPCR Testing

To detect *Brucella* spp. DNA in samples, the Taqman RT-PCR Bru Multi Assay was performed by targeting the *IS711* gene and using the primer sets and probes described by [11].

Each reaction mixture contained 6.25 µL of TaqMan Environmental MasterMix (Life Technologies, Brant, France), 0.75 µL of each primer (10 µM), 0.25 µL of probe (10 µM), and 8.5 µL of nuclease-free water, for a final volume of 25 µL. All reactions had a positive control that contained *B. abortus* DNA (1 ng/µL). Two negative controls were included: negative control of extraction (FBS), and negative control for amplification (H_2_O/nuclease-free). All DNA templates were tested in duplicate. An internal control using KoMa plasmid DNA [12] was also included. All amplifications were conducted with the program containing an initial step at 95 °C for 600 s, followed by 40 cycles of 15 s at 95 °C, and 60 s at 60 °C using the BioRad CFX96 cycler (BioRad, Singapore).

The data were analyzed using BioRad CFX Maestro Software. Samples with cycle threshold (Ct) values ≤ 37 were interpreted as positive.

All the positive samples were further analyzed by RT-PCR Bru Diff Assay to differentiate *B. abortus* and *B. melitensis* [11]. The amplification conditions were as described above.

All primers and probes used in this study (MOLBIOL, Berlin, Germany) were displayed in Table 1.

## 3. Results

### 3.1. Contamination Rates by Brucella spp.

In this study, 40 samples of cow’s milk, 102 samples of artisanal fresh cheese, and 58 ricotta samples were screened for the presence of *Brucella* spp. by qPCR through targeting the *IS711* fragment (Figure 2A). Out of the 200 samples, 150 (75%) were positive, and the Ct values ranged from 30 to 37. The negative controls were valid and no signal was detected. When looking at the results of each sample type, we found that 86.2% of ricotta, 69.6% of fresh cheese, and 72.5% of milk samples were positive. The *Brucella* spp. contamination rates in the different districts were found to be 94% in Tunis, 86% in Bizerte, 74% in Zaghouan, and 46% in Beja (Table 2).

### 3.2. Frequency of B. melitensis and B. abortus

All *Brucella* spp. positive samples were analyzed by qPCR for species identification (Figure 2B). Out of the 150 screened samples, 31.3% (47/150) were contaminated by *B. abortus*, while 5.3% (8/150) carried only *B. melitensis*. Most of the samples 49.3% (74/150) were double-contaminated with *B. abortus* and *B. melitensis*. While 14% (21/150) of the *Brucella* spp. positive samples were not identified either as *B. abortus* or *B. melitensis* (Table 2). The contamination rates of each sample type with *B. abortus* and/or *B. melitensis* are presented in Figure 3.

## 4. Discussion

Human brucellosis is a neglected zoonosis that remains a significant public health threat in urban and rural populations of endemic countries, particularly the MENA region [13]. In this area, the trade of fresh, unpasteurized milk and raw dairy products consumption is widespread. A large number of human brucellosis cases are attributed to the consumption of these products [14,15], which makes the control of *Brucella* spp. contamination in raw milk and unpasteurized derivatives a real challenge for food safety and consumer health protection.

In this study, we have assessed the occurrence of *Brucella* spp. contamination in raw milk and artisanal unpasteurized dairy products, which were sold in unorganized retail marketing points in northern Tunisia. The results showed high contamination rates; overall, 75% of the samples were positive for *Brucella* spp. by the IS711-qPCR. The contamination rates in the different sample types ranged from 86.2% in ricotta to 69.6% in fresh cheese. These high percentages might be linked to three main reasons: (i) the sensitivity of the detection method, (ii) the occurrence of the disease in milking females, and (iii) the cross contamination. Clearly, the percentage of *Brucella* spp. detection varies according to the sensitivity of the used method. The qPCR assays based on the use of the *B. melitensis* strain 16Minsertion sequence IS711, which is highly conserved in the genus *Brucella*, are very sensitive tools [11,16], thereby offering a very low detection limit of 10fg/reaction [16]. The analytical detection limit of the qPCR based on this target ranged from 1 to 6 genome copy/g in cheese [17]. Previous studies in India and Iran using PCR and quantitative real-time PCR showed high contamination rates in raw goat milk (88.8% and 45.5%, respectively) [18,19]. While using the milk ring test or ELISA, the contamination rates in cow′s milk ranged between 4.4 to 5.8% in India [20]. The efficiency of PCR, in comparison with the culture assay, was confirmed in many studies [21,22,23]. Marouf and collaborators have reported a variation of about 3.5 to 5 times between the prevalence estimated by qPCR and the culture test in dairy products [24]. Part of the difference between PCR and the culture test might be attributed to the error involved in the PCR technique discriminating between live and dead bacteria. In raw milk samples taken directly from the udder, the detected cells are likely viable, but in cheese, ricotta, or yogurt, the acidic conditions and NaCl concentration may affect *Brucella* viability [17]. To correct this bias, novel qPCR (PMA-PCR), using propidium monoazide (PMA) to inhibit the DNA amplification of dead bacterial cells, was developed to differentiate live from dead bacteria [25]. Moreover, real-time qPCR assays were developed to discriminate between virulent strains from those of vaccines [26].

On the other hand, the important levels of *Brucella* spp. contamination detected in this study could be associated with an increased occurrence of *Brucella* infections in flocks during the sampling period. Indeed, we have planned the sampling period from the early spring to late fall months, because this period coincides with an overproduction of milk and an increase in dairy products consumption, particularly during the month of Ramadan. Spring and fall correspond to the lambing seasons in Tunisia, and higher temperatures during this period increase husbandry activities, including the breeding, shearing, and commercialization of animals, which enhance the exposure of susceptible animals to different infectious diseases, notably brucellosis. The transmission and persistence of *Brucella* spp. may be enhanced in warmer conditions [27]. Data on the prevalence of animal brucellosis in Tunisia are scarce and fragmented, and the absence of specific and continuous surveillance programs and specialized diagnostic laboratories with adequate biosecurity conditions and sensitive diagnostic tools impeded the evaluation of the real and accurate extension of brucellosis infection in animals. The available data are mainly based on serological results, a retrospective study conducted by the CNVZ (Centre National de Veille Zoosanitaire) over a period of 14 years, has shown that the prevalence rates of infected ruminant herds in the country are very disparate and ranged from 0 to 70% of the flocks [28]. The study revealed that bovine brucellosis is mostly observed in the north and the southeast of the country, while the disease in small ruminant seems to be widespread in most districts. On the other hand, no seasonal peak was shown [28]. Another study conducted in the region of Sfax southeastern Tunisia, on 130 ruminant herds screened for brucellosis by qPCR, has shown that the prevalence of brucellosis was 55.6% in cattle and 21.8% in sheep [29]. The study revealed a significant association between vaginal and milk shedding of *Brucella* spp., which support our findings. Finally, cross-contamination might play a not insignificant role in the spread of *Brucella* spp. contamination. Indeed, the pooling of milk at the farm, with the local traders, and at the milk collection centers levels is a routine procedure in Tunisia. In this regard, contaminated milk from individual animals could likely contaminate the pool. In addition, the use of the same milking equipment, containers, and utensils without specific washing and sterilization measures increases the risk of cross-contamination. Second, at the dairy products vendors’ level, handling using the same knives to cut cheese and ricotta and the same pitcher to measure milk might increase the likelihood of cross-contamination between the different products. In this regard, the sources of contamination of dairy products in endemic regions are multiple, resulting in higher contamination rates.

The predominant detected *Brucella* species was *B. abortus* (31.3%), followed by *B. melitensis* (5.3%). We noted an important level of double-contaminated samples (49.3%) with *B. abortus* and *B. melitensis*, which confirmed that both species were circulating in the cattle population in Tunisia. The milk pooling remains the main source of this double-contamination. However, despite bovine brucellosis being typically caused by *B. abortus*, while ovine and caprine brucellosis is mainly caused by *B. melitensis*, cross-species infections are still possible [13]. Given the habit of raising small ruminants and cattle by small farmers, and the use of the Rev. 1 strain for vaccination in sheep and goats, the detection of *B. melitensis* could result from an infection of cattle with the Rev. 1 strain, as reported by other studies [30]. However, this might be confirmed by molecular typing of the detected strains. A part of the positive samples (14%) was not identified as either *B. abortus*, or *B. melitensis*, and this could be because of technical limits (low DNA concentration, most of Ct values are above 33) or due to the presence of other species, like *B. ovis*. Indeed, the contamination of milk could occur through two ways: the shedding of bacteria by the mammary glands of the infected females or from the contaminated environment during milking.

Despite this, the results are based on *Brucella* spp. DNA detection and don′t differentiate between live, damaged, or dead bacteria. The significant levels of *Brucella* spp. DNA in unpasteurized dairy products underscored the serious risk for consumers. Indeed, previous studies showed the ability of *Brucella* spp. cells to survive for weeks to months in acidic environments in dairy products [31,32]. It was also shown that the fat content of a dairy product can protect *Brucella* cells and enhance their survival capacity. Indeed, a previous study showed that viable bacteria decline more slowly in high-fat yogurt (five days at 3.5% of fat) than in low-fat yogurt (two days at 10% of fat) [33]. On the other hand, it was shown that prolonged storage of raw-milk cheese can affect the survival of pathogenic bacteria by the decrease of pH and the activity of lactic acid bacteria. Therefore, the ripening period, the acidic pH, and the potential production of antagonistic molecules by lactic acid bacteria are considered as major factors that influence the survival of *Brucella* in dairy products [17]. However, the bacteria can still survive under extreme alkaline (maximum pH 8.4) and acidic (minimum pH 4.1–4.5) conditions. Due to the high resistance of *Brucella* spp. to harsh environmental conditions, the European Commission (section IX) has prohibited the use of raw milk for human consumption and cheese production with a maturation period of less than two months if it does not come from official brucellosis-free holdings or from herds that are regularly checked for the disease under an approved control system [34].

Human and animal brucellosis are both notifiable diseases in Tunisia [35,36] and statistic data showed that the disease remains endemic, especially in rural areas [36,37]. The incidence of human brucellosis in Mediterranean countries is estimated to be 10/100,000 inhabitants, but it′s assumed that these statistics are underestimated. The World Health Organization argued that the incidence of human brucellosis in Maghreb countries is underestimated by 10- to 25-times of the actual value [38].

In Tunisia, human brucellosis presented recrudescent incidence over the years with national peaks of 1.28, 2.9, 4.35, and 8.94 per 100,000 inhabitants in 2003, 2011, 2015, and 2017, respectively. The southeast area maintains an endemic profile with a peak of 10.4 per 100,000 inhabitants in 2007. The highest annual incidence peaks were notified in 2007 (63.6), 2011 (48.9), and 2015 (30.8) per 100,000 inhabitants in the district of Gafsa (southwest) [36,39].

Most of the studies in Tunisia have reported a close association between human brucellosis and the consumption of raw milk and its derivatives. Indeed, 93.3% of clinical cases of human brucellosis (CHB) in the district of Gafsa were associated with the ingestion of raw milk and its derivatives [36]. While a retrospective study, conducted at the infectious diseases department of La Rabta Hospital in Tunis over a period of 17 years, revealed a percentage of 77% of neurobrucellosis cases attributed to foodborne contamination [40]. Another retrospective study conducted on patients with brucellosis in the department of Infectious Diseases in Sfax between 1990 and 2010, showed that 96.9% of patients have ingested unpasteurized milk or dairy products of infected cows [41]. In a recent study in the district of Tunis, 99.2% of the CHB cases reported also consumed raw dairy products [37]. The same study showed that 82.7% of cases were reported in summer and spring, with a peak in May. In correlation with other studies, the human brucellosis peaks were associated with high brucellosis incidence in animals [36].

Based on all these data, it seems clear that the eradication of human brucellosis should be taken within a “One Health” concept, which includes animal brucellosis control and food safety assurance.

## 5. Conclusions

Brucellosis infection through the consumption of dairy products is a serious hazard with great public health significance. The current study is the first in Tunisia to assess the occurrence of *Brucella* contamination in artisanal unpasteurized dairy products using qPCR. Although the use of PCR in the detection of *Brucella* spp. does not allow for the determination of the real risk associated with viable bacteria, it does allow an assessment of the risk of the presence of the bacteria in food. Our study provides evidence of the high contamination rates with *Brucella* DNA and the distribution of *Brucella* species in unpasteurized artisanal dairy products. Such findings should draw attention to the urgent need to revise the prophylactic measures and surveillance system and to implement meticulous and continuous control programs to limit and prevent brucellosis infection in ruminant herds. Fighting this major zoonosis must be conducted within an integrative “One Health” approach. Further research focusing on the isolation of *Brucella* spp. strains and molecular characterization are required to obtain a comprehensive understanding of brucellosis infection in animals and humans in Tunisia.

## Figures and Tables

**Figure 1 foods-11-02269-f001:**
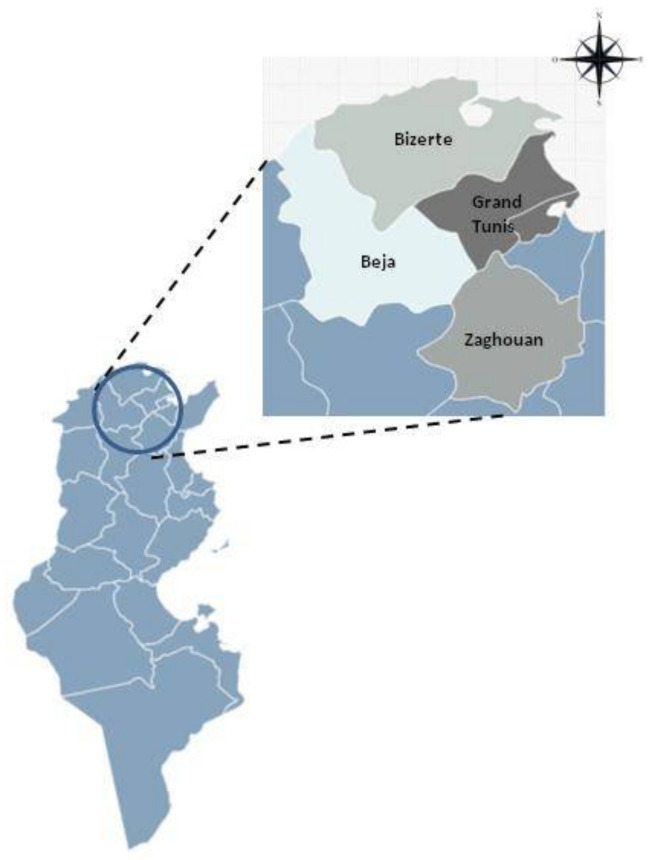
Location of the study area in Tunisia.

**Figure 2 foods-11-02269-f002:**
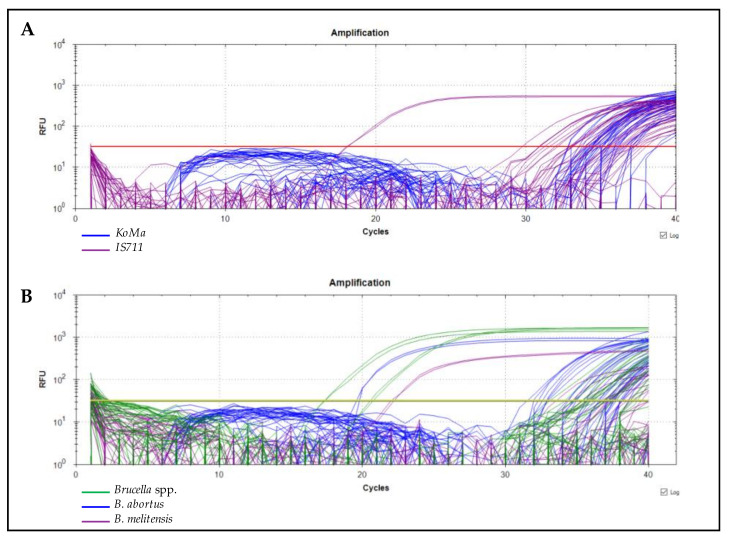
Amplification plot of the real-time PCR data. (**A**): *Brucella* spp. detection. (**B**): *B. abortus* and *B. melitensis* differentiation.

**Figure 3 foods-11-02269-f003:**
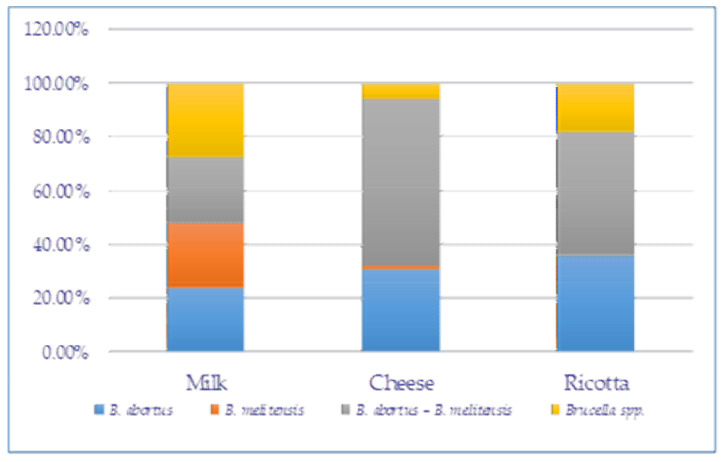
Distribution of *Brucella* species in positive samples.

**Table 1 foods-11-02269-t001:** qPCR primers and probes for *Brucella* spp. detection and species differentiation.

*Target*	*Primer/Probe*	*Sequence (5′-3′)*
** *Brucella* ** **spp. detection**		
*IS711*	Bru IS-F	GCCATCAGATTGAATGCTTTTTTAAC
	Bru IS-R	AACCAGATCATAGCGCATGCG
	Bru IS-TM	Cy5-CGCTGCGATGCGAGAAAACATTGACC-BHQ-2
*KoMa2*	KoMa-F	GGTGATGCCGCATTATTACTAGG
	KoMa-R	GGTATTAGCAGTCGCAGGCTT
	KoMa-TM	HEX-TTCTTGCTTGAGGATCTGTCGTGGATCG-BHQ-2
** *Brucella* ** **species differentiation**		
*Brucella spp.*	Bru-F	GCT CGG TTG CCA ATA TCA ATG C
	Bru-R	GGG TAA AGC GTC GCC AGA AG
	Bru-TM	FAM-AAA TCT TCC ACC TTG CCC TTG CCA TCA-BHQ-1
*B. abortus*	Bru ab-F	GCG GCT TTT CTA TCA CGG TAT TC
	Bru ab-R	CAT GCG CTA TCA CGG TAT TC
	Bru ab-TM	HEX-CGC TCA TGC TCG CCA GAC TTC AAT G-BHQ-1
*B. melitensis*	Bru mel-F	AAC AAG CGG CAC CCC TAA AA
	Bru mel-R	CAT GCG CTA TGA TCT GGT TAC
	Bru mel-TM	Cy5-CAG GAG TGT TTC GGC TCA GAA TAA TCC ACA-BHQ-2

**Table 2 foods-11-02269-t002:** Contamination rates with *B. abortus* and *B. melitensis*.

District	No. of Samples	No. of Positive Samples (%)	*Brucella* Species Differentiation			
			*B. abortus*	*B. melitensis*	Both Species *	*Brucella* Spp.
Beja	50	23 (46%)	4 (17.4%)	1 (4.3%)	9 (39.1%)	9 (39.1%)
Bizerte	50	43 (86%)	22 (51.2%)	-	21 (48.8)	-
Tunis	50	47 (94%)	8 (17%)	4 (8.5%)	28 (59.6%)	7 (14.9%)
Zaghouan	50	37 (74%)	13 (35.1%)	3 (8.1%)	16 (43.2%)	5 (13.5%)
Total	200	150 (75%)	47 (31.3%)	8 (5.3%)	74 (49.3%)	21 (14%)

Both species *: *B. abortus* + *B. melitensis*.

## Data Availability

All data generated for this study are contained within this article.

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
