# Peer review of "Brucella spp. Contamination in Artisanal Unpasteurized Dairy Products: An Emerging Foodborne Threat in Tunisia"

_foods, 2022, doi:10.3390/foods11152269_

Round 1
Reviewer 1 Report
Dear editor: Foods Journal
Thank you for contact me for reviewing the manuscript entitle”
Brucella contamination in artisanal unpasteurized dairy products: An emerging foodborne threat in Tunisia” manuscript ID: foods-1789262
After reviewing the manuscript, it was interesting and somewhat has anew novelty about the Brucella contamination in artisanal unpasteurized dairy products in Tunisia.
Some general comments and questions should be firstly answered:
1-Why the authors choose only 200 samples and not select equal number of samples from each food product to judged and compare between the results correctly
2-during the sample collection, it may contamination occur from the handlers or persons, not from the samples????
3- The table 1. Need to revision again with estimation according to total samples or total positive samples to give a nearly accurate status or % about brucella infection
4- The discussion section need more interpretation and explanation about the previous studies , also to clarified the reasons for high 5 of brucella infection in the current study
6- The conclusion section about the significance and the impact of the study should be addressed in points
7- Other comments are addressed in the reversed PDF manuscript.
8- The results may be improved by adding figures of the PCR and qPCR as supplementary figures.
Author Response
Dear reviewer,
Thank you very much for your time and effort to review this manuscript.
We revised this version according to your comments and suggestions. Please find attached our point-by-point response to your comments and questions.
We hope this version will meet your requirements.
Sincerely yours.
Awatef Béjaoui

Reviewer 2 Report
This study is interesting, and involves the presence of Brucella spp in dairy products collected in different geographical areas. However, it is necessary to follow the following recommendations:
Line 2: did you mean…Brucella spp. contamination…??
Line 4: it is necessary to remove the number in superscript format, when it is only an affiliation institution
Line 6: it is necessary to remove the number in superscript format, when it is only an affiliation institution
Line 15: did you mean…prevalence of Brucella spp. in 200…?
Line 19: ...prevalence of Brucella spp. was…
Line 23: …of Brucella spp. contamination…
Line 34: …diseases" [1]…
Line 64: …Brucella spp. contamination… ?
Line 69: According to the Microsoft Word Template from the authors guides, the subsection should not be written in bold text format
Line 70: could provide information regarding the climatic characteristics of the sampling area. How were the samples collected, in sterile containers? Was the transport carried out under controlled temperatures?
Line 71: samples can be separated into two climatic seasons (spring? summer? fall? winter?)
Line 89: According to the Microsoft Word Template from the authors guides, the subsection should not be written in bold text format
Line 91: specify equipment used to grind and homogenize samples (model, brand, city, country)
Line 97,98: insert space between both lines
Line 97: -20°C or -20 °C, like in line 108
Line 117: …and probes..?
Line 120: …Brucella spp. detection…
Line 128: …Brucella spp. differentiation….
Line 138: According to the Microsoft Word Template from the authors guides, the subsection should not be written in bold text format
Line 144: …The Brucella spp. contamination..
Line 150: According to the Microsoft Word Template from the authors guides, the subsection should not be written in bold text format
Line 151: …Brucella spp. positive samples,…
Line 176: …Brucella spp. differentiation…instead Brucella species Differentiation
Line 188: …Brucella spp. DNA..
Line 191: …Brucella spp. could..
Line 193: …Brucella spp. or Brucella spp. ?
Line 194: …Brucella spp. DNA..
Line 199: change [19,20,21]have by [19–21] have (see Microsoft Word Template)
Line 205: …Brucella spp. variability…
Line 213: peack or peak?
Line 218: …Brucella spp. was…
Line 239,249: …Brucella spp. or Brucella spp. ?
Line 241: Brucella spp. cells
Line 250: indicate the section number of the European Commission
Line 283: the conclusion must be modified; it must be associated with the results obtained in the study.
Line 307: Corbel, M.J.
Line 308: ... Available online: URL…
Line 310: Buttigieg, S.C.;
Line 310: according to what is indicated in the Microsoft Word Template, the title of the reference must be written in lowercase text format, except for the first letter of the first word
Line 311: according to what is indicated in the Microsoft Word Template, the abbreviated journal and the volume must be written in italic text format. Also, the year of publication must be in bold text format
Line 312: complete list of authors. The title of the reference must be written in lowercase text format, except for the first letter of the first word
Line 313: a dot must be added after each abbreviated word in the journal name of the reference
Line 315: the journal name of the reference should be abbreviated
Line 319,322,326,328: the title of the reference must be written in lowercase text format, except for the first letter of the first word
Line 319,322,326,346,350,364,376,378,380,382: the journal name of the reference should be abbreviated
Line 319: the volume must be written in italic text format
Line 322: correct author citation formatting, e.g. Franc, K.A.;….B.A. (make the same format correction for the list of authors in the lines 328,331,334,337, etc)
Line 326: correct author citation formatting, e.g. Hull, N.C.;….R.C.;…B.N.;…A.M.
Line 329,333,341,348,353,356,359,362,367,369,374,400,402: a dot must be added after each abbreviated word in the journal name of the reference
Line 331,334,337,347,352,358,373,375,399,401: complete list of authors
Line 331,334,337,340,345,347,349,352,355,358,361,363,366,368,373,375,378,380,382,390,399,401: The title of the reference must be written in lowercase text format, except for the first letter of the first word
Note 1: It is necessary to review the correct format for the references list in the Microsoft Word Template
Note 2: Although the research is interesting, and involves the presence of Brucella spp in dairy products collected in different geographical areas during the period from March to November 2019. It is important to separate these samples in at least two climatic seasons, in order to know if there were differences not only by region but by season of the year. This modification will further support the importance of research.
Author Response
Dear reviewer,
Thank you very much for your time and effort in reviewing our manuscript. We submit a revised version taking into account your comments and suggestions, we hope it will meet your requirements.
Please find attached our response to your comments.
Sincerely yours,
Awatef Béjaoui

Round 2
Reviewer 2 Report
All requested corrections were made